# Impact of Meteorological Changes on Particulate Matter and Aerosol Optical Depth in Seoul during the Months of June over Recent Decades

**Seohee H. Yang** [1], **Jaein I. Jeong** [2], **Rokjin J. Park** [2] and **Minjoong J. Kim** [1],*

[1]  Department of Environmental Engineering and Energy, Myongji University, Yongin, Gyeonggi 17058, Korea; seoheey0428@gmail.com

[2]  School of Earth and Environmental Sciences, Seoul National University, Seoul 08826, Korea; ss99@snu.ac.kr (J.I.J.); rjpark@snu.ac.kr (R.J.P.)

*  Correspondence: minjoongkim@mju.ac.kr; Tel.: +82-10-8779-8501

**Abstract:** The effects of meteorological changes on particulate matter with a diameter of 10 microns or less ($PM_{10}$, referred to as PM in this study) and aerosol optical depth (AOD) in Seoul were investigated using observational and modeling analysis. AOD satellite data were used, obtained from the Moderate Resolution Imaging Spectroradiometer (MODIS), and PM concentration data were used from in-situ observations. The Modern-Era Retrospective Analysis for Research and Applications (MERRA) and MERRA Version 2 (MERRA-2) were used for meteorological field analysis in modeling and observation data. The results from this investigation show that meteorological effects on PM and AOD were strong in the month of June, revealing a clear decreasing trend in recent decades. The investigation focused on the underlying mechanisms influencing the reduction in PM resulting from meteorological changes during the months of June. The results of this study reveal that decreases in atmospheric stability and humidity induced the aerosol change observed in recent decades. The changes in atmospheric stability and humidity are highly correlated with changes in the intensity of the East Asian summer monsoon (EASM). This suggests that the unstable and drying atmosphere by weakening of the EASM in recent decades has improved PM air quality in Seoul during the summer. The effects of atmospheric stability and humidity were also observed to vary depending on the aerosol species. Humidity only affects hydrophilic aerosols such as sulfate, nitrate, and ammonium, whereas atmospheric stability affects all species of aerosols, including carbonaceous aerosols.

**Keywords:** particulate matter (PM) variability; aerosol optical depth (AOD) variability; East Asia summer monsoon (EASM); atmospheric stability; climate change

## 1. Introduction

Particulate matter with a diameter of 10 microns or less ($PM_{10}$, referred to as PM in this study) is a representative indicator of air pollution that is known to affect human respiratory health [1] and cause lung cancer [2]. Seoul has developed rapidly since the 1960s and is characterized by a high population and dense vehicular traffic [3]. Along with its rapid economic and industrial development, Seoul concurrently exhibited a steep increase in PM concentration [4]. To improve air quality, the government of South Korea has implemented policies such as the Clean Air Conservation Act [5] and Special Act on the Improvement of Air Quality in Seoul Metropolitan Area [6], aimed at reducing the amount of emissions in order to decrease PM concentration in Seoul.

Owing to the enforcement of such government policies, PM concentration has decreased over the past few decades. However, episodes of high PM are still reported in Seoul [7]. Kim et al. [8] identified

the decline in wind speed as one of the reasons for this phenomenon. This indicates that PM levels are affected by both local emissions and meteorological conditions on a synoptic scale. Wind and air temperature gradient affect the horizontal and vertical ventilation of aerosols [9]. Wet deposition, hygroscopic growth, and a change of the aerosol formation by precipitation [10,11] and humidity [12,13] have also been reported as crucial factors affecting aerosol concentration.

Previous studies investigated the effect of meteorological factors on PM concentration in Seoul [8,11,14–19]. Most related studies have focused on the cold season (October–March) [7,17,18,20–22] because the episodes of high PM in Seoul occur most frequently in winter [17]. However, atmospheric aerosols show clear differences in summer and winter. The aerosol optical depth (AOD) is highest in summer, owing to the increasing hygroscopic growth of aerosols and formation of secondary aerosols, weak ventilation by meteorological effects, and transported pollution accumulated due to biomass burning from eastern China [23–27]. Constituents of aerosol also differ due to seasonal meteorological factors. In summer, sulfate is dominant by active photochemical processes due to strong solar flux, high temperature, and humidity, while nitrate and organic aerosol is highest in winter due to low temperature [28–32]. Although high aerosol concentration has been noted in cold season [33,34], aerosol changes in summer might also have a relationship with seasonal weather conditions; moreover, the weather system in East Asia is largely affected by the summer monsoon system. The mechanisms of meteorological effects on PM during summer might be different from those during winter. Therefore, the meteorological effect on PM and AOD in summer should be investigated.

Few studies have investigated the effect of the East Asian summer monsoon (EASM) on the PM level in East Asia and suggested that strong EASM decreases aerosol concentration in eastern China through precipitation and wind change [35–38]. These studies focus on the relation between EASM and PM, but not on the long-term changes in PM owing to variations in meteorological fields. However, recent changes in climate should affect the PM concentration in East Asia not only in winter but also in summer. Thus, in this study, we investigate the long-term variation of the meteorological effect on aerosol concentration in the Seoul Metropolitan Area during the month of June, when AOD is highest, using station observations, satellite data, reanalysis dataset, and model simulation results.

## 2. Data and Method

### 2.1. Data

We used the monthly observation data on the PM concentration and AOD to investigate the impact of meteorological change on PM in Seoul from 2003–2017. The PM data were obtained from 27 air quality monitoring stations constructed by the South Korea Ministry of Environment (MOE) in Seoul [39]. The observation stations are presented in Figure 1. Measurement of PM concentrations used the β-Ray absorption method, which observes the β-Ray absorption and dissipation when the β-Ray is passed through filter tape to collect dust. The PM was measured hourly for 24 h a day. The monthly PM concentration is calculated using arithmetic average of hourly PM.

We used the satellite dataset from the Moderate Resolution Imaging Spectroradiometer (MODIS) instrument onboard the National Aeronautics and Space Administration (NASA) Earth Observing System Aqua satellite to investigate the AOD changes in the Seoul Metropolitan Area, which is available for 2003–2017. Aqua passes at 13:30 local time. The data were extensively validated. The monthly averaged MODIS collection 6.1 level 3 products at 550 nm with 1° × 1° horizontal resolutions were used in this study [40].

Additionally, we employed a reanalysis dataset from the Modern-Era Retrospective Analysis for Research and Applications, Version 2 (MERRA-2) to estimate the long-term change in synoptic and climate patterns in (1980–2015) in East Asia [41]. MERRA-2 features approximately 0.5° × 0.625° resolution and 72 hybrid-eta levels from the surface to 0.1 hPa. MERRA-2 is computed on a latitude–longitude grid at the same spatial resolution using a 3-D variation data assimilation (3DVAR) algorithm based on the gridpoint statistical interpolation (GSI) with a 6-h update cycle using the

Goddard Earth Observing System Model, Version 5 (GEOS-5) with Atmospheric Data Assimilation System (ADAS), version 5.12.4. We used monthly averaged instantaneous 3-D collections of assimilated meteorological fields in single and pressure level, which is interpolated into 42 levels [42].

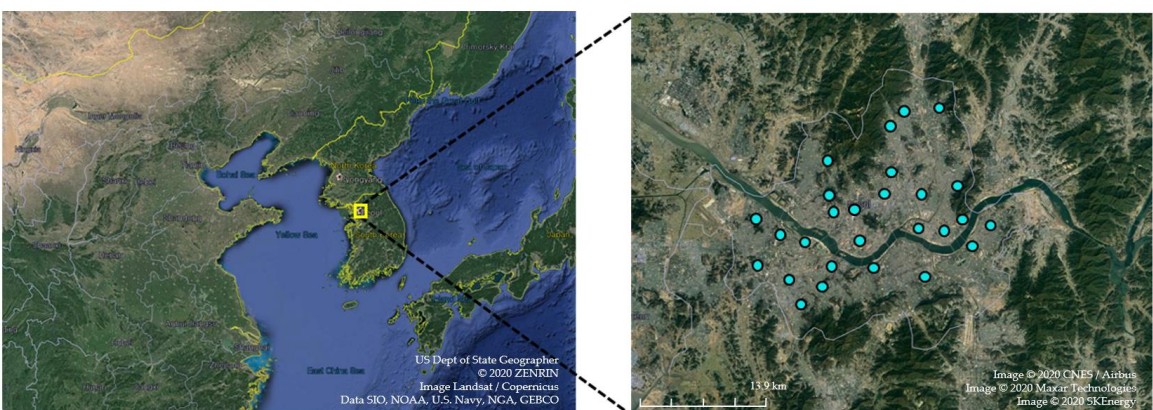

**Figure 1.** Monitoring stations of the surface PM concentration (blue dots) in Seoul. This map from Google Earth Pro. Compass of Seoul is [37.36°–37.74° N, 126.71°–127.34° E].

*2.2. Model Description*

We used the Goddard Earth Observing System (GEOS)-Chem v9-01-02 model, a global 3-D chemical transport model (CTM) for atmospheric composition [43], to simulate fully coupled oxidant-aerosols in the long-term in East Asia [44,45]. The GEOS-Chem model has been simulated in several air chemistry studies and evaluated against observation data in Asia, the United States, and Europe. Version 9-01-02 of GEOS-Chem uses assimilated meteorological data obtained from MERRA by NASA, which was completed in February 2016. The horizontal resolution of MERRA in the model was $0.5° \times 0.667°$, and the vertical level was divided into 72 hybrid sigma-pressure levels from the surface up to 0.1 hPa. The temporal resolutions of the MERRA were set at 3 h for wind, temperature, precipitation, and other 3-D variables, and at 1 h for surface variables. We interpolated the meteorological fields to a horizontal resolution from $0.5° \times 0.667°$ to $2° \times 2.5°$ and reduced the 72 vertical levels to 47 hybrid sigma-pressure vertical levels for computational expediency.

The model features more than 80 species and 300 reactions to allow for the detailed $O_3$-$NO_X$-hydrocarbon activity involved in aerosol chemistry. In the aerosol simulation, the model includes the aerosol thermodynamics of secondary aerosols, primary black carbon (BC), organic carbon (OC), and $H_2SO_4$-$HNO_3$-$NH_3$ [46–48]. To estimate $SO_4^{2-}$, $NO_3^-$, and $NH_4^+$ aerosols for the formation of $H_2SO_4$-$HNO_3$-$NH_3$ aerosols, ISORROPIA II, which is the thermodynamic equilibrium model, was applied in the model [49]. Fountoukis and Nenes [49] demonstrated that ISORROPIA II was appropriate for use in a large-scale air quality and atmospheric transport model because of this computational performance by optimizations in the activity coefficient calculation algorithm. The simulations of BC and OC have been described by Park et al. [47]. The dust entrainment and deposition (DEAD) mobilization scheme [50] and the Goddard Chemistry Aerosol Radiation and Transport (GOCART) model are used for calculating soil dust [51,52]. Sea salt in this simulation follows Alexander et al. [53]. Liu et al. [54] describe the wet deposition scheme in the model considering the effects of scavenging by convective updrafts, as well as rainout and washout by large-scale precipitation [55]. Dry deposition uses the size-segregated particle scheme by Zhang et al. [56]. The dry deposition scheme estimates particle dry deposition velocities using a function of particle size, density, and meteorological variables to accurately obtain realistic deposition velocities of sub-micron particles. We fixed anthropogenic emission in East Asia for the year 2006 as described by Zhang et al. [57] to isolate only the meteorological effect on aerosols after eliminating the effect of anthropogenic emissions in the long term. The anthropogenic emission dataset from Zhang et al. [57] was created for the Intercontinental Chemical Transport Experiment—Phase B (INTEX-B) project. INTEX-B emission is

designed to improve understanding of climate and aerosols conducted by NASA. INTEX-B emission inventory has been evaluated by numerous studies by comparing against aircraft observation, satellite data, and station observation [58–61].

### 2.3. Model Evaluation

We evaluate simulated aerosol concentration using observed near-surface aerosol concentration from the Acid Deposition Monitoring Network in East Asia (EANET) [62] and MOE. A substantial number of studies reported that simulated PM from the GEOS-Chem model captured observed PM under various conditions and observations [63–66]. We only focus on the evaluation of near-surface simulations of the PM concentration in June 2006 because anthropogenic emissions were fixed in 2006 in our simulation.

We used 17 out of a total of 49 EANET monitoring sites, all used for the PM concentration in East Asia (100° E–170° E, 0° N–60° N). These include 4, 10, and 3 EANET monitoring sites from China, Japan, and South Korea, respectively. We also used observed PM concentration from the MOE data, collected from seven metropolitan cities [39]. Figure 2A shows the comparison between observed and simulated PM concentrations on the surface in East Asia in the month of June. The average observed PM concentrations in South Korea, China, and Japan were 62.1, 60.0, and 20.8 µg m$^{-3}$, respectively. PM concentrations in Japan are generally much lower than those in China and South Korea because Japan is distant from the major anthropogenic emission sources in China [67]. The surface concentration in South Korea is higher than that in China because most of the observed stations in Korea are located in the megacity of South Korea. Among four EANET stations around China, the observed PM concentrations in Xiang Zhou and Hongwen, which are located in the southern coast in China, are 25.0 and 57.0 µg m$^{-3}$, respectively. On the contrary, PM concentrations observed in Jinyunshan and Weishuiyuan, which are located in inland urban cities, were 73.0 and 85.0 µg m$^{-3,}$ respectively. Concentrations in Jinyunshan and Weishuiyuan, which are influenced by anthropogenic emission and dust from the desert area, exhibit significantly higher concentrations of PM than the coastal area [68].

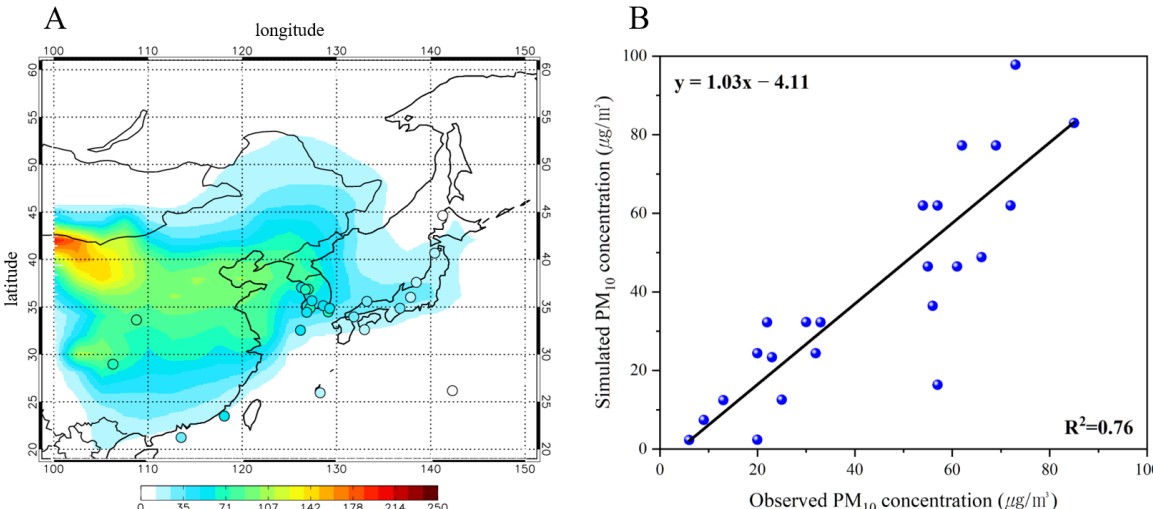

**Figure 2.** Comparison between simulated and observed surface PM concentration (µg m$^{-3}$) in June 2006. (**A**) Spatial distribution of surface PM concentration in the simulation over East Asia. Circles indicate the observed PM concentrations at Acid Deposition Monitoring Network in East Asia (EANET) and MOE monitoring sites. (**B**) Scatterplot of observed versus simulated PM concentrations at EANET sites. The black line indicates a linear trend of scatter plot.

Simulated PM concentrations are consistent with the observed PM concentrations. The simulated average of PM concentrations from the places where EANET and MOE monitoring stations are located yielded 59.57, 52.42, and 19.34 µg m$^{-3}$ in South Korea, China, and Japan, respectively. As with

the observed PM concentrations, higher concentrations are observed in South Korea than in China. In Xiang Zhou and Hongwen in the southern coast region in China, simulated PM concentrations were 16.37 and 12.55 µg m$^{-3}$, respectively. In Jinyunshan and Weishuiyuan, which are inland areas, simulated PM concentrations are 97.83 and 82.96 µg m$^{-3}$, respectively. Compared with the PM concentration in the inland area, the coastal area also yields significantly lower values in the simulation results. These results indicate that the model effectively reproduces the spatial distribution of observed PM concentration.

Figure 2B shows simulated versus observed monthly mean PM concentrations in all the EANET and MOE sites. The regression slopes between the observed and simulated values are close to unity, indicating no significant biases in the models (regression slope = 1.03 and intercept = −4.11). The coefficient of determination between the observed and simulated PM concentration is 0.76; it is statistically significant and has a confidence level of ≥99%. The consistency between the observed and modeled concentrations suggests the successful reproduction of aerosol simulation using GEOS-Chem in East Asia.

*2.4. Meteorological Indices*

We calculated the normalized near-surface wind speed index (WSI) and potential air temperature gradient index (ATGI) reported by Zou et al. [9] using the MERRA2 reanalysis data set to estimate synoptic atmospheric stability. The equations are as follows:

$$\text{WSI}^j_i = (\text{WS}^j_i - \text{WS}^j_{mean})/\text{WS}^j_{std} \tag{1}$$

$$\text{ATG} = \text{Air temperature }_{850\,hPa} - \text{Air temperature }_{10\,m\ \text{from surface}} \tag{2}$$

$$\text{ATGI}^j_i = (\text{ATG}^j_i - \text{ATG}^j_{mean})/\text{ATG}^j_{std} \tag{3}$$

In Equation (1), i represents the i-th year and j represents the grid point in the East Asia region. WSI$^j_i$ is a normalized WSI with no units. WS$^j_i$ refers to the monthly average wind speed in each month at 10 m height. WS$^j_{mean}$ represents the arithmetic mean of wind speed over 35 years, from 1980 to 2015 for the month of June. WS$^j_{std}$ is the standard deviation of wind speed from 1980 to 2015 for the month of June. Therefore, positive WSI indicates that wind speed is stronger than that of climatology at near-surface. The ATGI is estimated by normalizing the anomalies in the monthly average atmospheric temperature gradient (ATG) between 850 hPa and surface 10 m from 1980 to 2015. The ATG represents atmospheric stability at each grid point and is calculated by subtracting air temperature at 10 m from air temperature at 850 hPa. The unit for this variable is K. ATGI is calculated consistently with the WSI method. Thus, positive ATGI represents a more stable atmosphere below 850 hPa compared with that of climatology.

We also use the normalized near surface relative humidity index (HI) to investigate the effect of humidity on PM. HI is calculated like WSI. The unit for relative humidity is percent (%).

$$\text{HI}^j_i = (\text{Humidity}^j_i - \text{Humidity}^j_{mean})/\text{Humidity}^j_{std} \tag{4}$$

To estimate the effect of precipitation on PM, we used precipitation index (PREI) through a method similar to that of WSI calculation, and the unit of this variable is kg m$^{-2}$ s$^{-1}$.

$$\text{PREI}^j_i = (\text{Precipitation}^j_i - \text{Precipitation}^j_{mean})/\text{Precipitation}^j_{std} \tag{5}$$

We also applied the East Asian summer monsoon index (EASMI) to quantify the interannual variability of the EASM in South Korea. The EASMI is calculated by the flow of the 850 hPa lower jet

stream containing humid, warm air. We use the EASMI using the wind speed of 850 hPa in a specific area as suggested by Ha et al. [69]. EASMI is expressed as follows:

$$EASMI = (U_{850}^2 + V_{850}^2)^{1/2} \tag{6}$$

Here, $U_{850}$ represents the zonal wind component (m s$^{-1}$) at 850 hPa while $V_{850}$ represents the meridional wind component (m s$^{-1}$) at 850 hPa. $U_{850}$ and $V_{850}$ are averaged values over 32.5°–37.5° N, and 127.5°–147.5° E areas, respectively. There is an apparent positive correlation between this index and rainfall variability in South Korea [69]. Therefore, high EASMI indicates strong monsoon conditions prevailing near South Korea.

## 3. Results

Figure 3A shows the observed monthly averaged concentration and average annual percent change (AAPC) of aerosols for each month during 2003–2017 in Seoul. AAPC is calculated by linear yearly trend of normalized PM concentration and AOD in each months during 2003–2017. The monthly PM concentration in summer (June–August) is generally lower than those in other seasons owing to the wet scavenging of PM through monsoonal heavy rainfall [70]. The AAPC of PM yielded negative values for all the months due to the strengthening of air-quality policies. Notably, although the PM concentration decreases in all the months, the magnitude of AAPC of PM (Figure 3A; blue line) is the largest in June, showing −5.10% yr$^{-1}$ (2.58 µg m$^{-3}$ yr$^{-1}$), which is 2.5 times higher than in winter (−2.02% yr$^{-1}$).

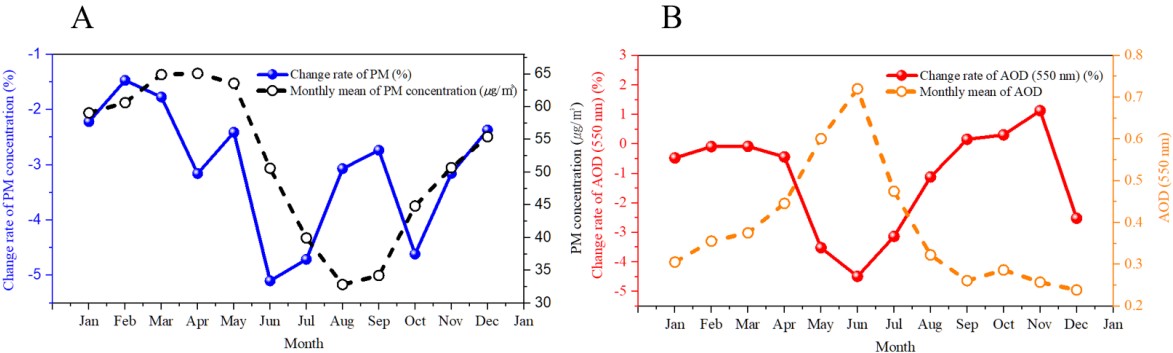

**Figure 3.** Monthly average and average annual percent change (AAPC) of surface PM concentration and aerosol optical depth (AOD) during 2003–2017 in Seoul. (**A**) Monthly average (dashed line) and AAPC (solid line) of surface PM concentration. (**B**) Monthly average (dashed line) and AAPC (solid line) of AOD. Unit of PM concentration is µg m$^{-3}$ and that of AAPC is percent (%). AOD is unitless.

Monthly AOD (Figure 3B; orange dashed line) is highest in the month of June (0.72), which is contrary to surface concentration. The high humidity and unstable atmosphere drive the high AOD through increasing secondary formation, hygroscopic growth of aerosols, and enhancing vertical dispersion of aerosols, regardless of the heavy rainfall in the month of June [23–27]. The average ground-level PM concentration and AOD in the month of June show opposite values due to vertically distributed aerosols. AAPC of AOD also yields the highest reduction in the month of June, showing −4.49% yr$^{-1}$ (0.032 yr$^{-1}$), which is consistent with that of surface PM concentration. Decrease in AAPC is highest in June for both PM and AOD for 14 y in Seoul. The high reduction of AAPC in the month of June may be attributable to changes in the meteorological condition rather than changes in emissions, because air-quality policies are implemented year-round, not focused on a specific month.

Therefore, we focus on the roles of meteorological change in aerosols in the month of June. We first calculate the linear trends of meteorological indices in Seoul in the month of June for 2003–2017 to estimate the long-term change in each meteorological variable (Figure 4). WSI does not show a clear linear trend (Figure S1), whereas ATGI, HI, and PREI present clear deceasing trends, showing −0.16 yr$^{-1}$, −0.16 yr$^{-1}$, and −0.11 yr$^{-1}$, respectively, in Seoul over the 14 years. These linear trends

may be an indication of the possible correlation between the decrease in PM in recent decades and reductions in stability, humidity, and precipitation.

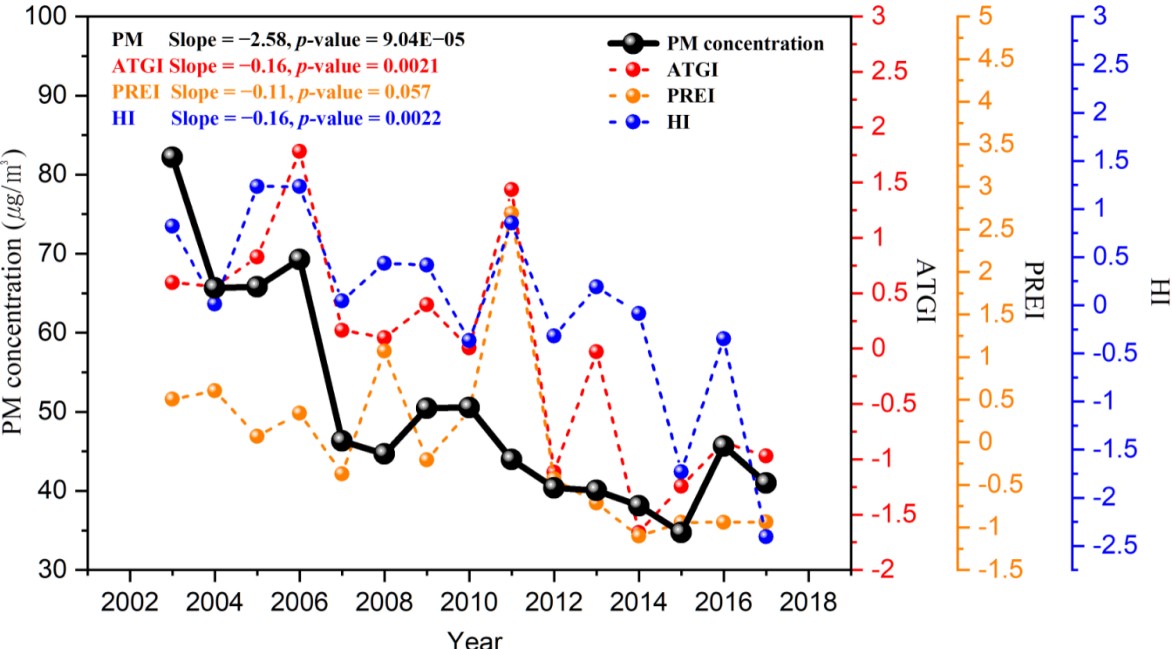

**Figure 4.** Time-series of PM concentration and normalized meteorological indices in June for 2003–2017 in Seoul. All indices are calculated from the MERRA-2 reanalysis dataset. Unit of PM concentration is μg m$^{-3}$, and meteorological indices are unitless.

We also conducted a regression analysis to quantify the effect of each meteorological factor on the observed change in aerosols during the month of June. Figure 5 exhibits regressed observed PM and AOD in Seoul against meteorological indices at each grid in the month of June during 2003–2017. These figures indicate the sensitivity of the PM and AOD in Seoul response to meteorological indices at each grid. WSI near Seoul is positively correlated with the observed PM concentration in Seoul; however, the correlations between WSI and PM are not significant in South Korea at a 95% confidence level (Figure 5A). WSI also shows no clear correlation with observed AOD in South Korea (Figure 5B). These results imply that the ventilation caused by wind could not have a significant effect on the concentration of aerosols, which differs from those reported by previous studies that focused on winter [9,17]. The different mechanisms may be attributable to weak wind speed in the month of June compared to winter [24]. Weak relationship between PM and WSI might also suggest that the potential role of long-range transport of PM in June is weaker than that in winter season. Note that, generally, westerly wind over Korea and Eastern China is weak in summer, owing to expansion of the North Pacific High [71]. However, our analysis only considers a limited effect of long-range transport on PM and AOD. Previous studies suggest that change in biomass-burning emissions over Eastern China significantly affect AOD in Seoul [24,27]; however, biomass-burning emissions over Eastern China have increased in recent years [72]. Increasing biomass-burning emissions might compensate the effects of wind on PM and AOD. Our analysis cannot reflect the effect of these changes.

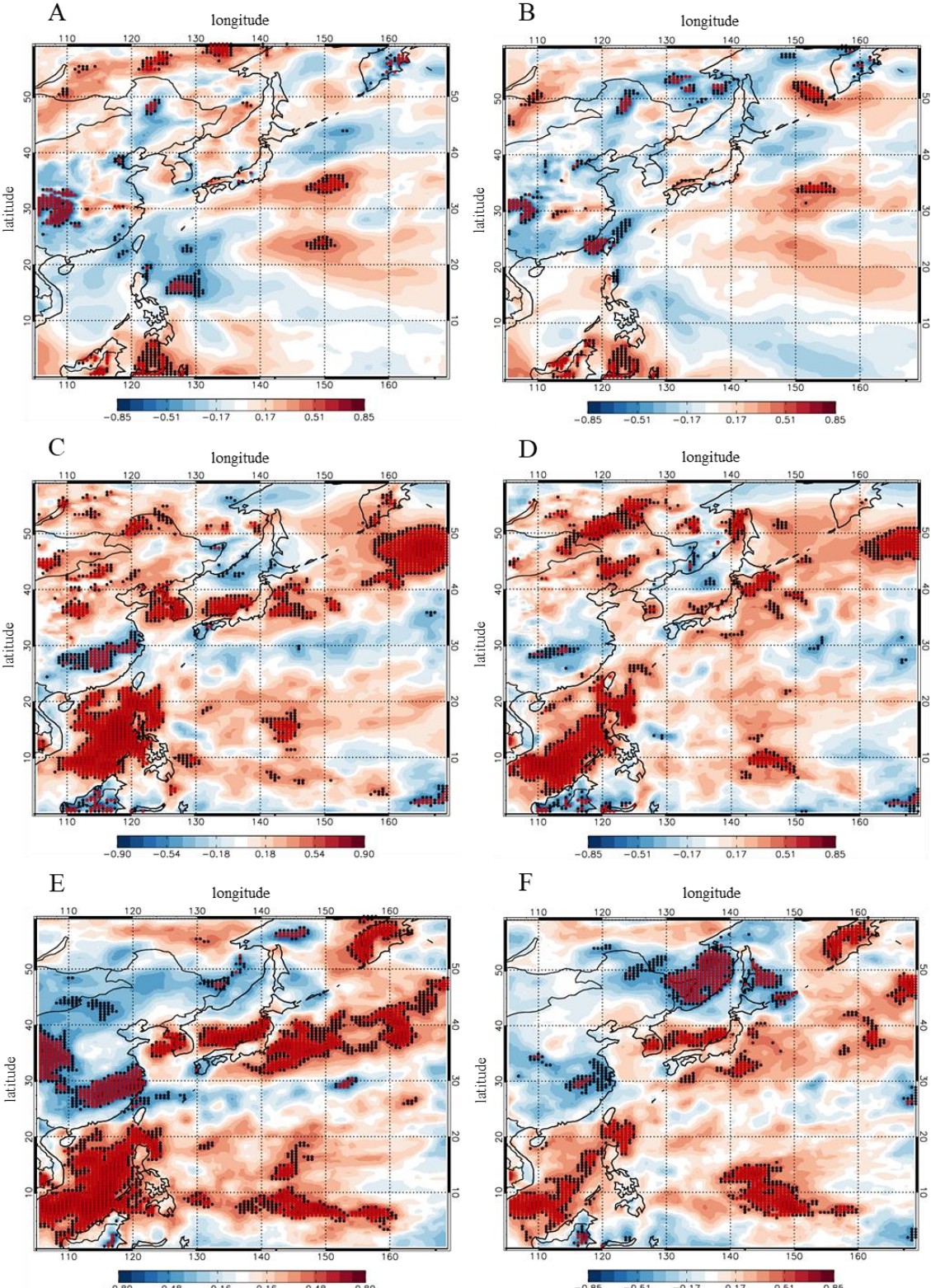

**Figure 5.** Regressed normalized PM and AOD in Seoul against meteorological indices at each grid in the month of June for 2003–2017. (**A**,**B**) are regressed PM concentration and AOD in Seoul against WSI at each grid. (**C**,**D**) are regressed PM concentration and AOD in Seoul against ATGI at each grid. (**E**,**F**) are regressed PM concentration and AOD in Seoul against HI at each grid. Black dots indicate statistical significance at the more than 90% level and red dots indicate statistical significance at the more than 95% level. Normalized PM concentration and AOD are from station observation and MODIS. Meteorological indices are calculated from MERRA-2 reanalysis dataset. All variables are unitless.

In contrast, ATGI clearly exhibits a positive linear correlation with both PM concentration and AOD in South Korea at the 95% confidence level (Figure 5C,D). The sensitivity of PM concentration and AOD in Seoul response to ATGI also shows the highest value near Seoul. A previous study insists that change in vertical atmospheric stability is an essential meteorological factor affecting air pollutants, especially in winter in East Asia [9]. Our results suggest that the change in aerosols in South Korea is closely associated with the changes in vertical atmospheric stability in South Korea not only in winter but also in summer. The decreasing atmospheric stability in recent decades as well as the positive correlation with aerosols may have led to improvements in the air quality in South Korea through the dispersion of aerosols, as suggested in previous studies. Moreover, the EASM during summer results in an unstable atmosphere [73]. Large changes in atmospheric stability resulting from change in EASM may be crucial for PM air quality in South Korea, implying an underlying mechanism of EASM on PM air quality.

We also found that humidity is a major meteorological variable for PM variation in South Korea in the month of June. Regressed PM concentration and AOD against humidity indicate a strong positive linear correlation at the 95% confidence level (Figure 5E,F). Humidity is more sensitive in summer due to the inflow of humid air by EASM [73]. Humid air transported from Northwestern Pacific promotes the formation of secondary aerosols such as sulfate, nitrate, ammonium, and organic aerosols as well as the light extinction of aerosols by hygroscopic growth of hydrophilic aerosols [13,74], thereby increasing surface PM concentration and AOD. The decreasing humidity over recent decades, which is positively correlated with aerosols, may result in the improvement of PM air quality, similarly to ATGI.

Precipitation and aerosol concentration are typically negatively correlated owing to the wet scavenging processes [10,11]; however, the regressed PM concentration and AOD in Seoul against precipitation exhibit a significant positive correlation in South Korea, which is contrary to general intuition (Figure S2). Previous studies state that short and light rains in the urban area increase aerosol concentration because aerosol increase by hygroscopic growth is greater than the removal of aerosol through wet scavenging [75–77]. An observed campaign in East Asia also reported that short heavy rains could not sufficiently reduce aerosol concentration through wet scavenging [78]. Thus, these results show that the effect of increased humidity on aerosol is greater than that of wet scavenging, which is consistent with findings in previous studies.

The effects of meteorological variables on aerosols are related with changes in large-scale EASM strength, which is a major climatological variation in East Asia as earlier explained. Previous studies reported that the large-scale variation of EASM affects the PM level through change in the atmospheric condition in East Asia during summer [35–38]. EASM's linear trend declined during 2003–2017, when atmospheric stability and humidity decreased (Figure 6). In this study, we infer that the weakening of EASM over the past few decades has induced improvements in PM air quality in South Korea.

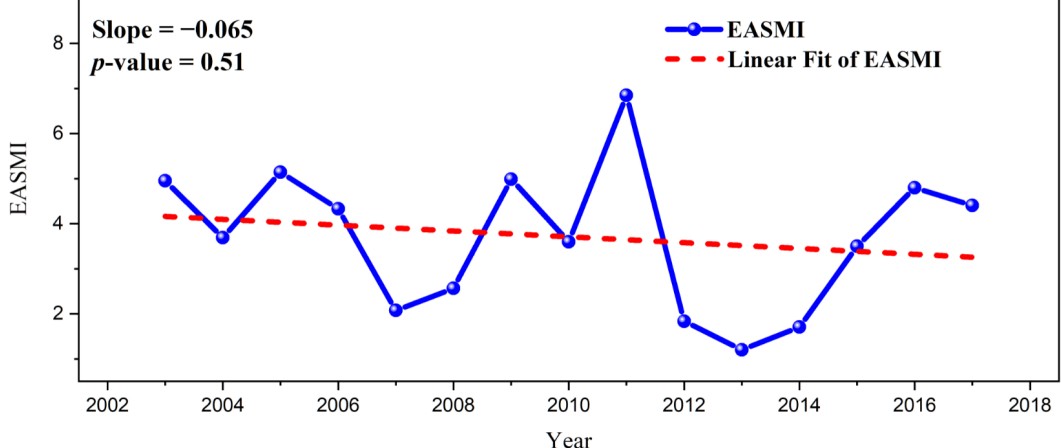

**Figure 6.** Time-series of East Asian summer monsoon index (EASMI) from the MERRA2 reanalysis dataset in the month of June for 2003–2017. Red line is the linear trend of time-series. This variable is unitless.

To determine the underlying mechanism of EASM and meteorological variables, we calculated the linear regressed atmospheric variables against EASM strength. Figure 7 indicates regressed ATGI and HI against EASMI for the month of June for 1980–2017. Both regressed ATGI and HI have positive signals at a 99% confidence level with EASMI in South Korea. These results indicate that a weakened EASM results in an unstable and dry atmospheric condition, which may suppress the degradation of aerosols owing to the unfavorable atmospheric condition of stagnant aerosol growth and formation. To reveal the mechanism underlying EASM and atmospheric stability, we calculate regressed air temperature at 850 hPa and 10 m against EASMI (Figure S3). The regressed temperature at two layers shows the opposite signal. Regressed air temperature at 850 hPa was positively correlated with EASMI in South Korea at a 99% confidence level, whereas no relationship was evident between temperature at 10 m and EASMI. Thus, weakened EASM hinders the transportation of warm air at 850 hPa from the south, thereby destabilizing the atmospheric condition in South Korea. We also found that EASM is positively related with humidity at the 99% confidence level in South Korea, indicating that weakened EASM inhibits the transportation of humid air into South Korea. Regressed wind against the EASMI shows the southerly wind, indicating that strong EASM transports wet and warm air from Southeast Asia (Figure S4).

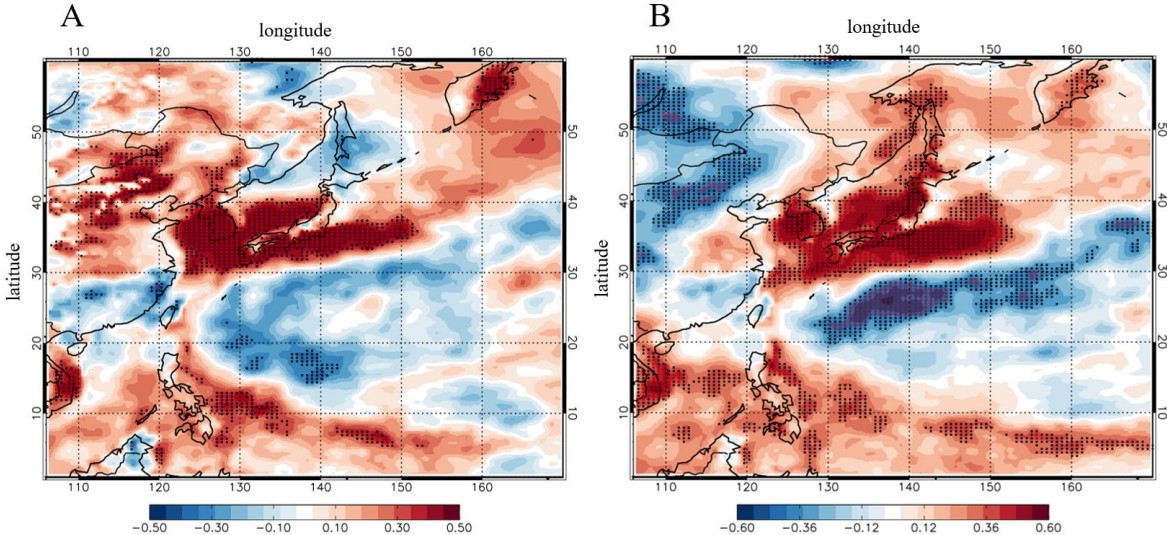

**Figure 7.** Regressed (**A**) ATGI and (**B**) HI at each grid against normalized EASMI for 1980–2017 during the month of June. Black dots indicate a statistical significance level of more than 95% and red dots indicate a statistical significance level of more than 99%. All indices are calculated from MERRA-2 reanalysis dataset. All regressed data are unitless.

The decrease in PM concentration in recent decades may have affected the reduction of emissions over the same period. To examine the effect of climate changes alone on PM air quality, we conducted a model simulation using GEOS-Chem during 1980–2015 assuming no changes in anthropogenic PM precursor emissions from the present-day values in 2006. The linear trend of simulated PM concentration was negative for 1980–2015 during the month of June ($-0.20$ µg m$^{-3}$ yr$^{-1}$; Figure 8A). However, the decreasing trend of simulated PM is 12 times lower than that of observed PM for 2003–2017 ($-2.58$ µg m$^{-3}$ yr$^{-1}$). This result implies that the effect of climate changes alone on PM is much weaker than that of the emission changes, since anthropogenic emissions from South Korea and China have been drastically reduced in recent years [8,79]. ATGI and HI also showed a decreasing trend during 1980–2015 by $-0.041$ yr$^{-1}$ and $-0.026$ yr$^{-1}$ in the model, similar with that from the MERRA-2 reanalysis dataset. We also found that decreasing trends of ATGI and HI are much smaller than those of the MERRA-2 reanalysis data for 2003–2017 (both are $-0.16$ yr$^{-1}$), similar to that of PM. These results might suggest that the weak trend of PM in the model is owing to weak changes in humidity and stability during a longer period.

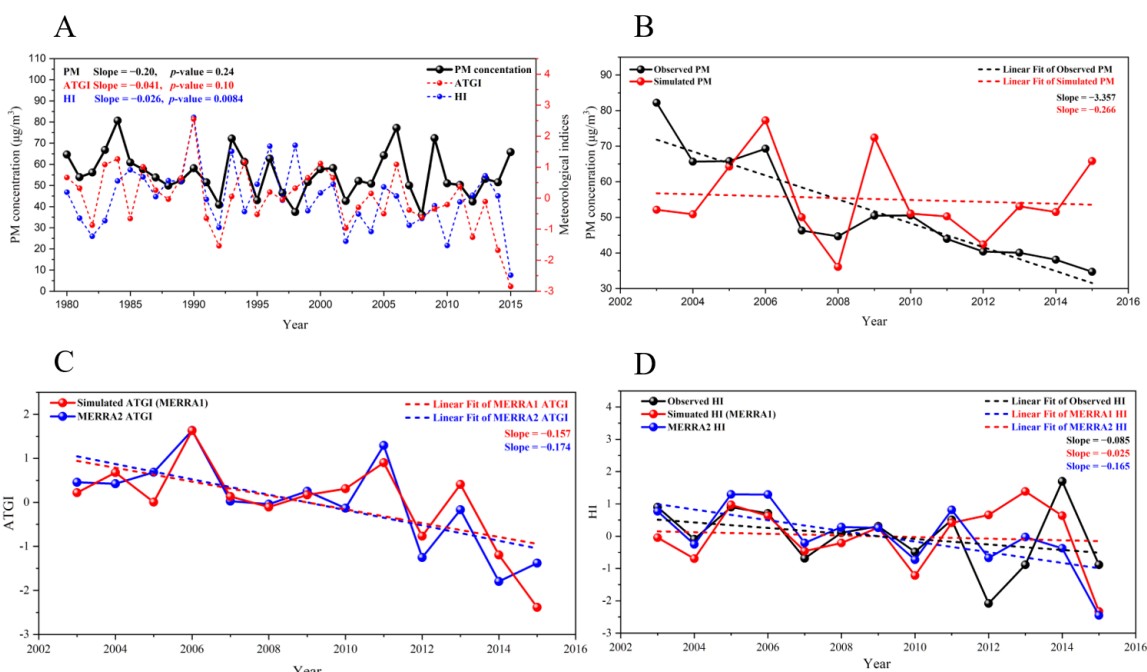

**Figure 8.** (**A**) Time series of the PM concentration, ATGI, and HI using simulated dataset (PM concentration: black solid line; ATGI: red dashed line; HI: blue dashed line) for 1980–2015 in Seoul. (**B**) Time series of the PM concentration using observed (black solid line) and simulated (red solid line) datasets for 2003–2015 in Seoul. Dashed lines indicate the trends of station-observed PM (black dashed line) and simulated PM (red dashed line). (**C**) Time series of ATGI using MERRA reanalysis with $2° \times 2.5°$ resolution (meteorological field dataset in model; red solid line), and MERRA-2 with $0.5° \times 0.625°$ resolution (blue solid line) for 2003–2015 in Seoul. Dashed lines indicate MERRA ATGI (red dashed line) and MERRA-2 ATGI (blue dashed line). (**D**) Time series of HI using station observed humidity (black solid line), MERRA (red solid line), and MERRA-2 (blue solid line) for 2003–2015 in Seoul. Dashed lines indicate the linear regression lines between two variables. All indices are unitless, and the unit of PM concentration is $\mu g \, m^{-3}$.

We compare PM in the model with observed PM during 2003–2015, which is an overlapped period between model and observation. The linear trend of the simulated PM during 2003–2015 was 12 times lower than the observed PM trend ($-0.27 \, yr^{-1}$ and $-3.36 \, yr^{-1}$, respectively; Figure 8B). We also compare humidity and stability from MERRA reanalysis with $2° \times 2.5°$ horizontal resolution, which is the input of GEOS-Chem, against those from MERRA-2 reanalysis with $0.5° \times 0.625°$ horizontal resolution. Linear trends ($-0.16 \, yr^{-1}$, $-0.17 \, yr^{-1}$, for MERRA and MERRA-2 respectively) and variations of ATGI from the two datasets are close to each other and show significantly similar patterns (correlation coefficient = 0.88, *p*-value = $7.34 \times 10^{-5}$; Figure 8C). However, the linear trend of HI in MERRA ($-0.025 \, yr^{-1}$) is seven times lower than that of MERRA-2 ($-0.17 \, yr^{-1}$), although the analysis period is reduced (Figure 8D). This result suggests that MERRA reanalysis with $2° \times 2.5°$ resolution has large discrepancy compared with MERRA-2, especially for humidity, owing to its coarse resolution; $2° \times 2.5°$ resolution cannot distinguish the effect of sea because Seoul is not far from the Yellow sea (~40 km). We compared relative humidity using station observations in Seoul. The linear trend of station observations ($-0.085 \, yr^{-1}$) is 3.4 times higher than that from MERRA, indicating that the linear trend of MERRA is low at the grid of Seoul, owing to its coarse resolution. We conclude that the decreasing trend of PM might be underestimated in the model due to the lower humidity trend in Seoul. Despite large discrepancies in magnitude of linear trends, variables in the model exhibited constant tendencies in the observation and reanalysis dataset. Therefore, we more focused on the relationship between meteorological variables and PM using regression analysis than the physical magnitude of the linear trend, because of the limitation of input data of the model due to its coarse resolution.

Figure 9 displays the map of regressed PM concentration in Seoul against ATGI and HI at each grid during 1980–2015 in the month of June in the simulation. PM concentration has a significant positive correlation and high sensitivity against ATGI and HI in the model near South Korea at a 99% confidence level, which is consistent with results from reanalysis data sets and in-situ observation. The spatial pattern of the regressed map is slightly different from the observation and MERRA-2 reanalysis dataset result, which shows the westward shift of positive peak in the yellow sea. This difference between the simulated and observed dataset might be attributable to the coarse resolution of the model; however, simulated results are generally consistent with those of the observed dataset, which indicates that the relationship between meteorological variables and PM air quality is robust and distinct from the effect of emission change. The regressed PM against ATGI in the Seoul is 0.33 in the model during 1980–2015, which is 50% of those observed during 2003–2017 (0.67; Figure 5C). The regressed PM against HI in the Seoul is 0.15 in the model, which is four times lower than that of observed PM (0.59). However, the averaged value of regressed PM against HI including downwind region (123.75°–128.75° E, 35°–39° N; yellow dashed box in Figure 9B) is 0.51, which is 86% of those from observed. These results suggest that the sensitivity of PM response to ATGI and HI in the model is similar with that of observed PM, despite a reduction of simulated PM being much lower than that of observed PM. Considering that the model used fixed anthropogenic emission, meteorology changes explain 25–86% of the total PM reduction following differences of the regression slope of PM against meteorological indices in the simulation and observation. These results imply that although the simulation has limitations owing to its coarse resolution, the decreasing trend of PM is significantly affected not only by emission changes, but also by changes in atmospheric stability and humidity. However, we did not conduct simulations with changing emissions, and simulation was only evaluated against the observation during 2006. Therefore, our analysis has explicit limitations and uncertainty due to differences between the real world and the model.

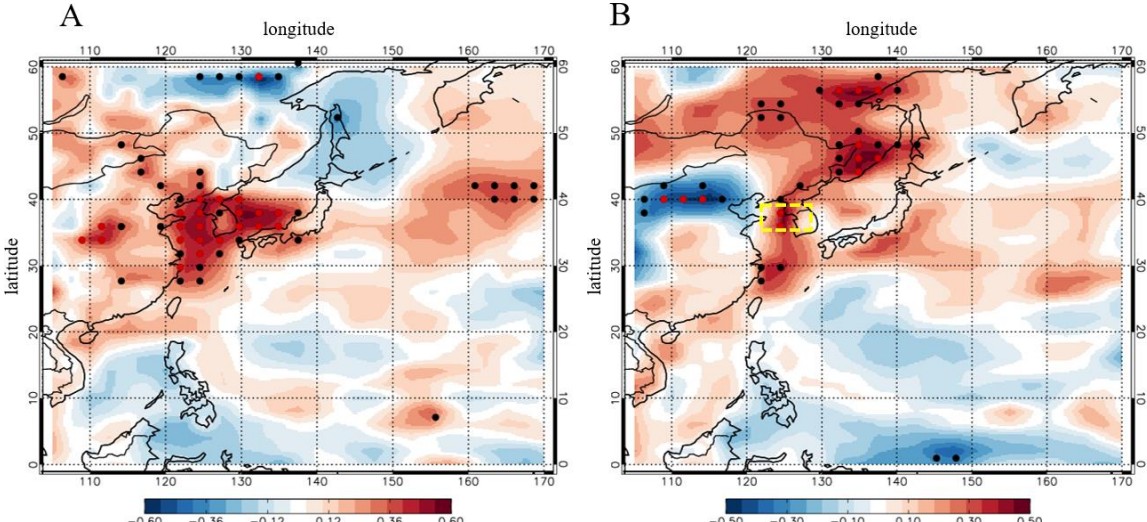

**Figure 9.** Regressed normalized PM in Seoul against (**A**) ATGI and (**B**) HI at each grid during the month of June from 1980–2015. Black dots indicate a statistical significance level of more than 95%, and red dots indicate a statistical significance level of more than 99%. All data is from GEOS-Chem simulation and MERRA reanalysis. All regressed data are unitless.

We additionally examined the meteorological effects on each aerosol species. We found that ATGI was positively correlated with all species, including both secondary inorganic aerosols (SIA; sulfate, nitrate, and ammonium aerosols) and organic aerosols, which include organic and black carbon (Figure 10A,B). This result shows that changes in stability affect all the aerosol species through changes in ventilation. However, HI is only correlated with SIA, suggesting that the humidity affects hydrophilic aerosols via formation of secondary aerosols (Figure 10C,D). Note that SIA aerosols

are regarded as hydrophilic in the model, whereas 50–80% of carbonaceous aerosols are emitted as hydrophobic [47,80]. Therefore, the decreasing trends of SIA are significantly larger than those of black and organic carbon owing to the effect of humidity on hydrophilic aerosols (Figure S5). These results indicate that atmospheric stability and humidity have different effects on each type of aerosol.

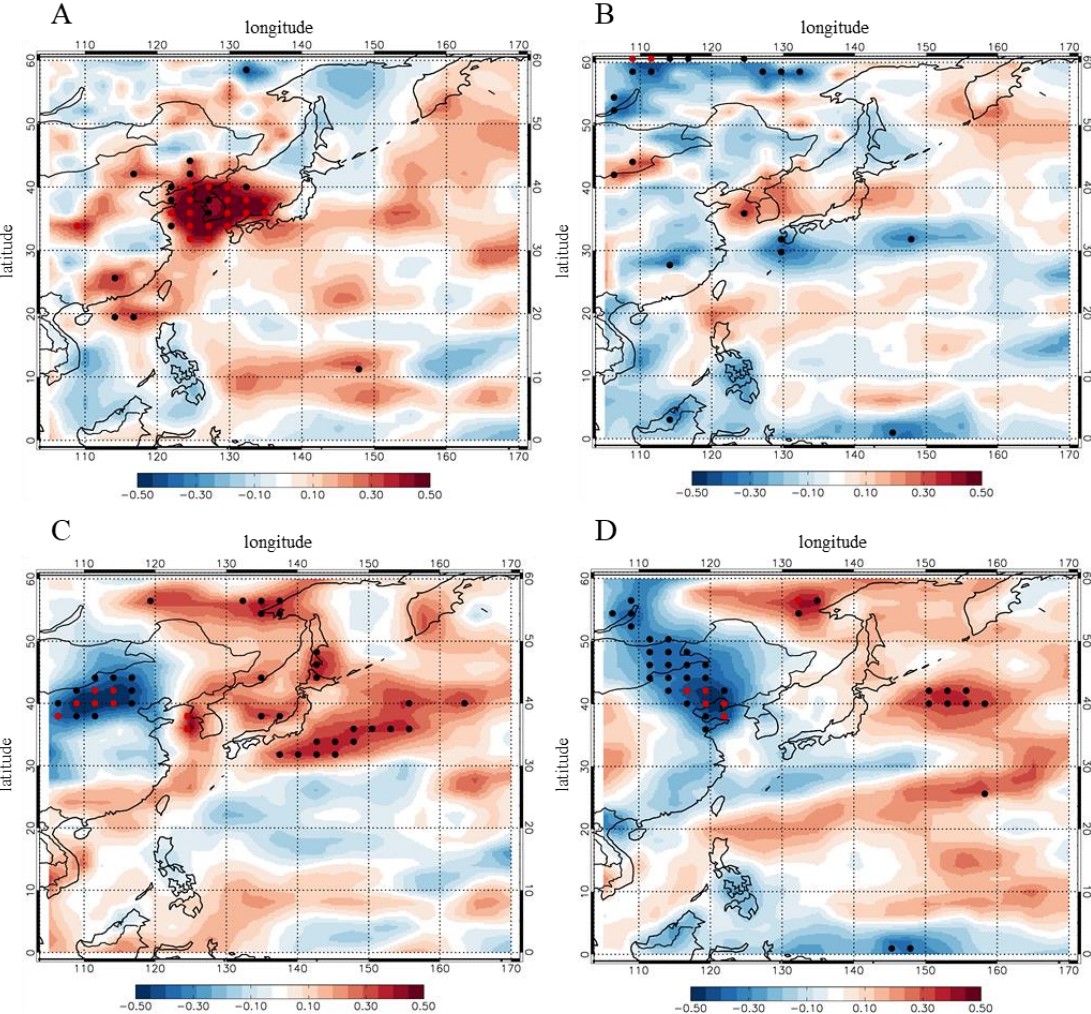

**Figure 10.** Regressed normalized (**A**) SIA and (**B**) carbonaceous aerosols in Seoul against ATGI at each grid during the month of June for 1980–2015. Regressed normalized (**C**) SIA and (**D**) carbonaceous aerosols in Seoul against HI at each grid during the month of June for 1980–2015. Black dots indicate a statistical significance level of more than 95%, and red dots indicate a statistical significance level of more than 99%. All data is from GEOS-Chem simulation and MERRA reanalysis. All regressed data are unitless.

We investigated the relationship between EASMI and meteorological factors in the model by conducting regression analysis to analyze the effect of large-scale climate variation on PM air quality (Figure 11). Despite the decreasing trend of EASMI during 1980–2015 ($-0.006$ yr$^{-1}$) being much weaker than that of MERRA-2 reanalysis during 2003–2017 (Figure S6A), regressed ATGI and HI against EASMI have clear positive signals at the 99% confidence level in South Korea, which is consistent with the results from the MERRA-2 reanalysis dataset. Moreover, the EASMI in the MERRA reanalysis are well matched with those of MERRA-2 for 2003–2017 (Figure S6B). These results indicate that the weakening of EASM in recent decades has induced the improvement of PM air quality during the month of June through the increased instability and dryness of the atmospheric condition, even though we excluded the effect of emission change.

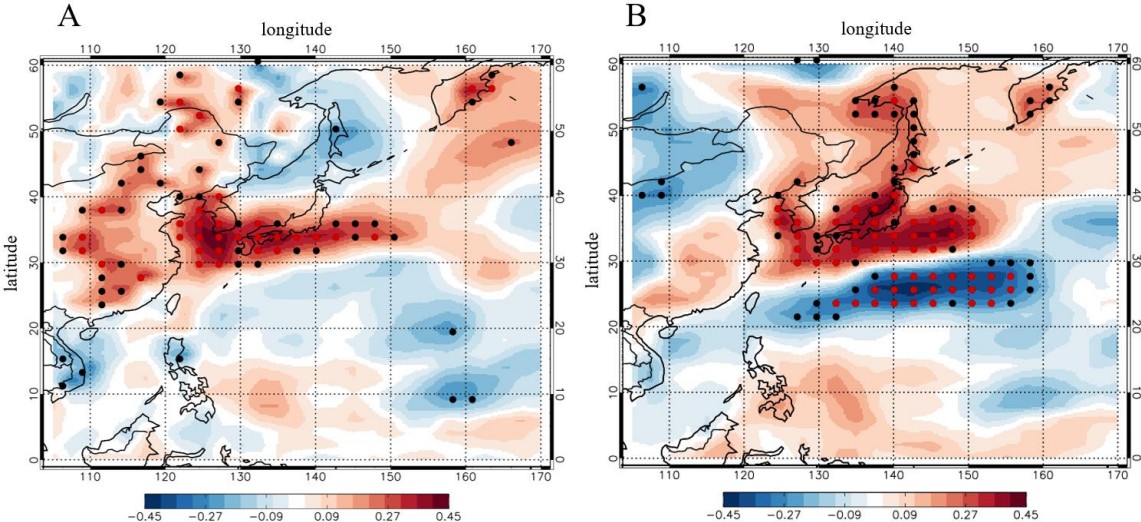

**Figure 11.** Regressed (**A**) ATGI and (**B**) HI at each grid against EASMI during the month of June from 1980–2015. Black dots indicate a statistical significance level of more than 95%, and red dots indicate a statistical significance level of more than 99%. All data is from MERRA reanalysis. All regressed data are unitless.

Figure 12 displays the linear slope of regressed normalized PM concentration at each grid against EASMI for 1980–2015. We found that EASM and aerosol concentration have a negative relationship in China but have a positive relationship in South Korea. Our results suggest that the mechanism between the EASM and PM level is different in China and South Korea. Several studies have reported that the EASM and aerosol concentration are negatively correlated in the whole of East Asia, which is contrary to our results [37,38]. The different sign between the EASM and PM level in South Korea may be related to different vertical resolutions of the meteorological dataset in the model. We used a MERRA dataset with 47 vertical levels, whereas previous studies used the GEOS of the NASA GMAO version 4, which features 30 vertical levels and a significantly coarser vertical resolution than those of MERRA, especially near the surface. Considering that our results are consistent with in-situ observation and reanalysis datasets, previous studies cannot capture the mechanism between PM and EASM in South Korea owing to their coarse vertical resolutions.

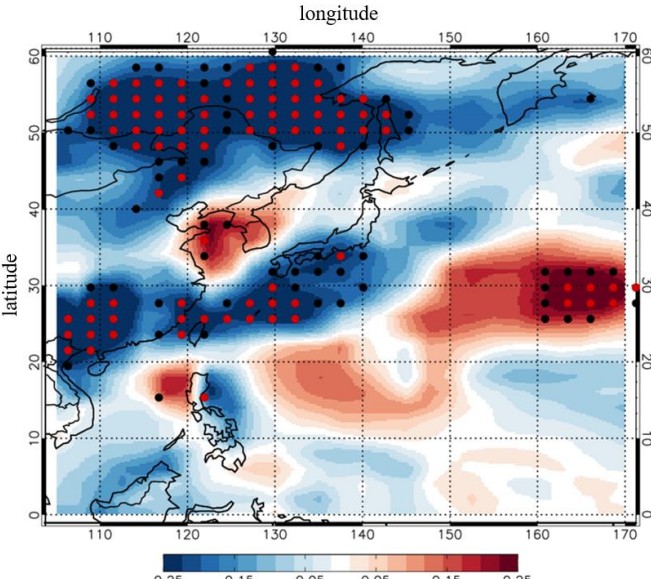

**Figure 12.** Regressed normalized PM concentration at each grid against EASMI during the month of June for 1980–2015. Black dots indicate a statistical significance level of more than 90%, and red dots indicate a statistical significance of more than 95%. All data is from GEOS-Chem simulation and MERRA reanalysis. All regressed data are unitless.

## 4. Conclusions

　　PM concentration in Seoul has been decreasing owing to the government's emission reduction policies since the 2000s. Aerosols exhibit a sharp, and the largest, reduction in the month of June. The highest reduction of aerosols in the month of June may be influenced by changes in meteorological factors because air-quality policies are implemented throughout the year. We investigated the effect of meteorological factors on aerosol in the month of June over recent decades using in-situ observations, satellite data, reanalysis data, and modeling.

　　We found that the observed PM concentration and AOD in Seoul are correlated with increasing unstable and dry conditions in the month of June during 2003–2017. Increases in unstable and dry atmospheric conditions are attributed to the weakened EASM, which inhibits the hot and humid southwesterly wind. We also conducted simulation to isolate the effect of climate change on PM air quality, assuming no changes in anthropogenic PM precursor emissions. Simulated trends of meteorological variables and PM concentration were consistent with those of observed PM, despite the magnitude of linear trends in simulation being lower than those of the observation owing to the coarse horizontal resolution of the model. We found that unstable and dry atmospheres exhibit a significant positive correlation with aerosol concentration in Seoul, indicating the robustness of the mechanism of aerosol reduction in the observation.

　　Relationships among PM, meteorological variables, and strength of EASM in the simulation also correspond with those observed, indicating that the strength change in EASM in recent decades has improved the PM air quality in Seoul. We also investigated the meteorological effects on each aerosol species in Seoul. The effects of humidity and atmospheric stability on aerosol varied depending on the type of aerosol. The changes in stability affect all the aerosol species, whereas humidity only affects SIA via promotion of aerosol generation.

　　Our results suggest that long-term meteorological changes in the month of June might accelerate the reduction of aerosols owing to emission control policies in Seoul. We also found that long-term meteorological changes were closely correlated to the strength of EASM. We should pay attention to future changes in large-scale variability including EASM to determine future air quality policies.

**Supplementary Materials:** The following are available online at http://www.mdpi.com/2073-4433/11/12/1282/s1, Figure S1, Time-series of WSI June for 2003–2017 in Seoul. This index is calculated from the MERRA-2 reanalysis dataset. WSI is unitless. Figure S2, Regressed normalized PM and AOD in Seoul against meteorological indices at each grid in the month of June for 2003–2017. (A) and (B) are regressed PM concentration and AOD in Seoul against PREI at each grid. Black dots indicate statistical significance at the 90% level and red dots indicate statistical significance at the 95% level. Normalized PM concentration and AOD are from station observation and MODIS. Meteorological indices are calculated from MERRA-2 reanalysis dataset. All variables are unitless. Figure S3, Regressed normalized temperature at (A) 850 hPa and (B) 10 m for each grid against normalized EASMI in the month of June for 1980–2017. Black dots indicate statistical significance at the 95% level and red dots indicate statistical significance at the 99% level. Normalized EASMI and temperature are calculated from MERRA-2 reanalysis dataset. All variables are unitless. Figure S4, Regressed normalized wind vector (zonal wind; meridional wind) and normalized surface pressure (shading) at each grid against EASMI in the month of June for 1980–2017. Expressed wind vectors indicate statistical significance at the 95% level. Unit of wind vector is m s$^{-1}$. Normalized EASMI, wind vector, and surface pressure are calculated from MERRA-2 reanalysis dataset. Figure S5, (A) Time series of the secondary inorganic aerosols (SIA), (B) organic carbon (OC), and (C) black carbon (BC) using MERRA reanalysis dataset at Seoul in the month of June for 1980–2015. Red lines indicate linear regression lines of each index. The unit of aerosol concentration is μg m$^{-3}$. Figure S6, Time-series of EASMI at Seoul in the month of June. (A) is time series of the EASMI using the MERRA reanalysis dataset at Seoul in the month of June for 1980–2015. Red line is the linear regression line of the EASMI. (B) is time series of the EASMI using the MERRA reanalysis dataset at Seoul in the month of June for 2003–2015 (red solid line). Blue line is the time-series of EASMI using MERRA-2 reanalysis during the month of June for 2003–2017. All variables are unitless.

**Author Contributions:** Conceptualization, investigation, formal analysis, methodology, S.H.Y., M.J.K.; Software, J.I.J. and R.J.P.; Writing—original draft preparation, S.H.Y. and M.J.K.; Writing—review and editing, S.H.Y., M.J.K., J.I.J. and R.J.P. All authors have read and agreed to the published version of the manuscript.

**Funding:** This work was funded by the National Research Foundation of Korea (NRF) grant funded by the Korea government (MSIT) (No. NRF-2020R1C1C1008898).

**Conflicts of Interest:** The authors declare no conflict of interest.

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
