# Peer review of "Impact of Meteorological Changes on Particulate Matter and Aerosol Optical Depth in Seoul during the Months of June over Recent Decades"

_atmosphere, doi:10.3390/atmos11121282_

Round 1
Reviewer 1 Report
L372 (C)->(D)
Author Response
We appreciate for your considerable help in developing our manuscript.
L372 (C)->(D)
We corrected.
Reviewer 2 Report
The revised manuscript fully meet the reviewer's points.
It can be entered the publication stage.
Author Response
We appreciate for your considerable help in developing our manuscript.
Reviewer 3 Report
The manuscripts was much improved. However, there are still some confusions that authors can make them clear. Please see below comments.
- The paper needs to claim that this study ignored the aerosol emission variation during the study period of 2003-2015. The model uses anthropogenic emission inventory in the year of 2006 and keeps it not changed for the simulations of PM and PM species in the year of 2003-2015. This study only validated the simulated PM with the observation data in June 2006 in Fig.2. But as shown in Fig.8(b), different variation trends of PM between the model simulated data and the observation data are indicated, and the model simulation product do not clearly indicate the decrease trend while the observation data clearly indicate the decrease trend from 2003 to 2015. In China, the PM emission inventory was changed dramatically during 2003-2015 which may affect the PM over South Korea through long-range transport. Such model simulations of PM and its species in this study probably make big bias during 2003-2015.
- Some definitions or descriptions on the weak or strong atmospheric conditions (Stability, humidity, precipitation) are not clearly given. For instance, for Eq.(3) and Line 185-193, air temperature gradient index (ATGI) is defined to quantify atmospheric stability. However, authors did not clarify at what level the ATGI represents strong stability or weak stability or instability of the atmosphere. Is there any threshold or positive/negative value to classify either strong or weak atmospheric stability?
Same concerns for WSI, HI, PREI and EASMI in Eq.(1)-(6), at what levels (threshold) they represent weaker or stronger wind speed, humidity or precipitation.
- Line 403 about SIA (Secondary inorganic aerosols), how the model separate the primary inorganic aerosols from the secondary inorganic aerosols?
- Line 223-226, why these meteorological conditions (high humidity and low wind speed) increase AOD in June, but not increase ground-level PM? Under such stagnant and humid weather, both AOD and ground PM should follow similar increase trends.
- Fig.12, how to define the Regressed normalized PM concentration at each grid against EASMI? (linear slope or correlation coefficient?)
- In the Abstract, please clarify the “PM” for PM10 or PM2.5?
Author Response
Please see the attached file as below

Round 2
Reviewer 3 Report
The revised manuscript has been much improved. I recommend it to be accepted for the publication.
This manuscript is a resubmission of an earlier submission. The following is a list of the peer review reports and author responses from that submission.
Round 1
Reviewer 1 Report
In this manuscript, the authors investigated the influence of meteorological changes on aerosol concentration. Understanding effects of meteorological changes on aerosol properties is important. The information in this manuscript is helpful for our better understanding of air quality. However, I think current form of this manuscript is not acceptable to publish and I request you change several parts of this manuscript. Detailed comments are followed.
Major comments:
The authors described that the decrease in humidity could lead to prevention of hygroscopic growth of aerosols, and eventually decrease PM. However, unlike AOD, PM10 is known to indicate the mass concentration of dried aerosols to minimize effects of humidity. According to table 1, the change rate in PM concentration by humidity change was higher than that in AOD. I think that more detailed descriptions are needed for these results. Aqueous phase reactions may have contributed to the increase in PM concentration under high humidity conditions.
The authors described that meteorological changes have a large effect on the reduction of aerosol concentration. However, the decreasing rate of PM was very lower than the observed trend (P11L316) when the author assumed no changes in anthropogenic PM precursor emissions from the present-day values in 2006. If so, it seems that the effect of the meteorological change was insignificant than that of PM precursor emission reduction. For these claims to be convincing, more detailed explanations are needed.
According to Fig 9, 11, 12, ATGI has a positive correlation with PM, and EASMI also has a positive correlation with ATGI in East China. But why do PM and EASMI show a negative correlation in East China?
Minor comments:
Since Aqua passes over this region at 13:30 LT, AOD data at that time would be used. How about the PM data, only daytime or 24-hour?
P3L127: Full name of EANET was not described before this sentence.
P5L186: The authors mentioned that the averaged value of EASMI in this area was used in this manuscript. In Fig. 7, however, the regression analysis was carried out at each grid point. Which one did you use, the average value or the value at each grid point?
P6L214: PREI also showed a decreasing trend.
Fig. 4: It would be better to add the time-series of PM in June for 2003-2017, similar to Fig. 4.
Fig. 9: Which one did you use, PM concentration in Seoul or PM concentration at each grid?
P14L368: EASMI in Fig. S5 does not show a declining trend.
Reviewer 2 Report
This paper investigated aerosol trends in June using the observed and simulated PM10 in terms of meteorological variables. The authors tried to elucidate the decrease of PM in June in East Asia because of the weakened EASM which inhibits the hot and humid southwesterly wind. The manuscript had an important scientific finding, but some points will be clarified during the revision process. The ‘results and discussion’ are too long to follow their logic, thus the manuscript needs to be concise. Please divide it into a sub-section for readability. And please clarify the data source coming from (e.g., observed or simulated)
Minor comments:
- L48: Please add more recent paper.
- L49-L52: The reference is also too old (2007) and recently, the winter smog events are receiving more attention due to their high concentration (Kim et 2017, Ghim et al., 2019). Moreover, secondary inorganic ions during smog events in winter are much higher than those in summer according to the above-mentioned references. Please provide more relevant papers for proving the summertime high concentration or rephrase the sentence.
- L117: What emission inventory did you use? Are there any specific reasons to use the 2006 base?
- Why was the PM concentration compared only in 2006? Although the PM was simulated based on the 2006 emission inventory, the comparison should be conducted until 2017 because the simulated PM used trend analysis. Since this comparison has had a lot of discussion in Korea, at least a Korean PM comparison should be done.
- L183: Please add the more detailed description of ‘East Asian summer monsoon index’. Readers might be confused with wind speed.
- Please add the p-value for the slopes in Figure 4, 6, and 8.
- L301-310: I’m afraid to mention the specific number of PM decrease because those two variables are not strongly linked because R value was moderate.
- L307-308: It is hard to believe the PM decreased due to the decreasing trend of RH because the PM was measured at 40% RH condition using the heater to exclude the bias from hygroscopic aerosols. Thus, the reason of decreasing RH is much suitable for explaining the decreasing AOD only.
- L318-320: please add the slope from the same time scale with observed PM (2003-2017). And how about the correlation between yearly observed PM10 and the simulated one.
- L326-333: Where did the PM data come from? MOE or MERRA or GEOS-CHEM?
- L341: observed one? Isn’t it MERRA data?
Reviewer 3 Report
This study performed GEOS-Chem simulations to investigate the relationship between meteorological factors and air quality in Seoul. This paper has some interesting findings and its topic fits the scope of the Atmosphere. In general, this paper is clearly written and well structured, although I suggest the authors further edit the language.
Comments:
In the introduction, the authors mentioned high PM episodes, while the results showed that there are decreases in concentrations induced by meteorology. It might help if the authors can specify the conditions of high PM episodes.
It seems that the model resolution is coarse. Will adopting higher resolution change the results?
430: this sentence is not clearly written. Please revise “accelerate reduction owing to…”
Reviewer 4 Report
The paper presents the meteorological impact on PM and aerosol optical depth in Seoul in June of 2003–2017. The observation and model data are analyzed. It focus on the meteorological effects on the PM10 variation trend, but ignore the major role of anthropogenic emissions reduction and long-range transport of pollutants during the study time period. The results heavily depend on the model data used, but there are more accurate observation data available for both meteorological and AOD data. The paper is generally organized and written well, but there are some confusions or missing information on the results and discussions. Please see the comments below.
- The paper discussed and pointed out the humidity and atmospheric stability impact on PM concentration, but ignore the fact that the EPA measured PM are dry mass of PM.
In the study time period in 2003-2017, the anthropogenic emissions, particularly for PM, have been reduced, this may play a major role on the PM10 reduction.
In addition, authors did not discuss the potential role of long-range transport of PM in June, which might be associated with the agricultural biomass burning events in China and partially affect the PM level in South Korea.
- The model data for the meteorological and AOD data are overwhelmingly used to assess their temporal variation trends. But there are extensive observational and more accurate data available, including the winds, potential temperature and humidity at the weather stations, as well as the radiosonde profile data.
In addition, there are extensive observed AOD data available from the AERONET-sites in South Korea, they are more accurate that the MODIS AOD data. The below link shows the AERONET site map: https://aeronet.gsfc.nasa.gov/cgi-bin/draw_map_display_aod_v3
Overall, the observation data are more accurate than the model data to verify their variation trends in South Korea.
- Fig.3, How do you calculate these change rates of PM and AOD?
the high AOD and lowest surface PM10 in summer might be also due to aloft aerosol plumes that are associated the agricultural biomass burning and dust aerosols in June.
- Line 256-259, The EPA usually measures the dry PM mass concentration that are not affected by humidity.
- Line 313-314, “The decrease in PM concentration in recent decades may have affected the reduction of emissions over the same period.” I think that the reduction of emissions plays a major role against the meteorological effect discussed in this paper.
- Line 355-356, Confusion to this sentence, “Note that SIA aerosols are mostly hydrophilic whereas, carbonaceous aerosols are significantly larger hydrophobic aerosols than SIA aerosols.” Where is this conclusion from?
- Line 305-308, how do you calculate these percentage contribution from the atmospheric stability and humidity?
Line 377-380, “PM changes are calculated by multiplying the regression slope by the amount of change in meteorological variables”. The uncertainty of the regression slope might be very large because the regression correlation coefficients seem low.
- Line 430-431, “Our results suggest that long-term meteorological changes in the month of June accelerate the 430 PM and AOD decrease owing to emission control policies in Seoul.”
This study did not demonstrate the influence of the emission control.
- In the Title, “air quality” is not necessary.
In the Abstract, please point out the data sources that were used such as the model for meteorological parameters and AOD from the satellite product.